# Effect of COVID-19 Lockdown on Children’s Eating Behaviours: A Longitudinal Study

**DOI:** 10.3390/children9071078

**Published:** 2022-07-20

**Authors:** Giuseppina Rosaria Umano, Giulia Rondinelli, Giulio Rivetti, Angela Klain, Francesca Aiello, Michele Miraglia del Giudice, Fabio Decimo, Alfonso Papparella, Emanuele Miraglia del Giudice

**Affiliations:** Department of the Woman, the Child, and General and Specialized Surgery, University of Campania Luigi Vanvitelli, 80138 Naples, Italy; giuliarondinelli90@gmail.com (G.R.); giuliorivetti94@gmail.com (G.R.); klainangela95@gmail.com (A.K.); francesca.aiello.alagi@gmail.com (F.A.); michele.miragliadelgiudice@unicampania.it (M.M.d.G.); fabio.decimo@unicampania.it (F.D.); alfonso.papparella@unicampania.it (A.P.); emanuele.miragliadelgiudice@unicampania.it (E.M.d.G.)

**Keywords:** eating behaviour, COVID-19, pediatric obesity, lifestyle, questionnaire

## Abstract

The COVID-19 pandemic has had a huge impact on children’s lifestyle and eating behaviour, resulting in an increase of obesity prevalence. The CEBQ (Children’s Eating Behaviour Questionnaire) is a validate questionnaire that investigates children’s eating behaviour. Knowing the psychological consequences of daily routine disruption during lockdown, we evaluated the changes in eating behaviours in a paediatric cohort before and during the lockdown period through the evaluation of the Italian version of the CEBQ. We prospectively enrolled children attending the pediatric clinic of the University of Campania ‘Luigi Vanvitelli’. All parents answered the parent-report version of the CEBQ before lockdown containment. During lockdown, the second survey was carried out by telephone call. The study sample included 69 children. Food responsiveness and emotional overeating subscales showed higher scores during lockdown compared to data before lockdown (*p* = 0.009 and *p* = 0.001, respectively). Conversely, desire to drink and satiety responsiveness showed lower scores at follow-up (*p* = 0.04 and *p* = 0.0001, respectively). No differences were observed for slowness in eating and enjoyment of food. Delta changes were higher in normal-weight children compared to children with obesity (*p* = 0.02). Our results confirm that containment measures during the COVID-19 pandemic have acted as triggers on certain eating behaviors that mostly predispose to an obesogenic manner.

## 1. Introduction

The SARSCoV-2 pandemic has been responsible for lifestyle changes in the world population including children and adolescents. All countries around the world adopted different strategies to contain the spread of the virus. In Italy, the containment measures were very stringent, particularly during the first pandemic wave where a total lockdown, with the closure of schools and all social activities, occurred. The sedentary lifestyle caused by the interruption of the working, recreational, and educational activities determined an increase in new cases of obesity and the worsening of the pre-existing ones [1,2]. It has been found that children’s snacking and consumption of energy-dense foods, sedentary behaviour, and screen time have increased while physical activity has declined [3].

The general panic at the beginning of the pandemic has led several families to buy massively long-life foods that are typically ultra-processed and calorie-dense. In addition, the pandemic also had socio-economic consequences: many people lost jobs, leading to financial constraints with a consequent increase in the consumption of readily available, dense, high-calorie foods and sugary drinks. There was also an increase of the intake of potato chips, red meat, and sugary drinks [4]. This affected the susceptibility to weight gain, especially considering the drastic reduction of everyday physical activity and the increase of screen time [5,6]. It has been estimated, on average, over 5 h a day are spent on computers, TV, and video games, both for school activities and for social communication, and for the entertainment of children and adolescents, exacerbating sedentary habits and increasing the risks of anxiety, depression, inattention, and sleep quality disruption [7,8,9]. According to Di Renzo et al., in a cohort of 602 children in Italy, during the COVID-19 lockdown, almost half of the respondents felt anxious due to their eating habits, consumption of comfort foods, and inclination to increase food intake in order to feel better [10]. The COVID-19 pandemic exerts disproportionate burden, especially on low-income children and families, magnifying their vulnerability to both food insecurity and pediatric obesity [11]. In this context, children’s eating behavior should be carefully evaluated to quantitatively estimate changes in eating habits. To this aim, the Children’s Eating Behavior Questionnaire (CEBQ) is a validated parent report questionnaire that has been extensively used in research and clinical practice [12,13]. CEBQ subscales scores have been associated with obesity risk and have been used successfully in different populations of children to measure associations between eating behaviors and relative weight of children [14,15,16,17]. It has also been used in other studies, for example, to compare appetite preferences in children of thin and obese parents, to study continuity and stability in children’s eating behavior during infancy, and the relationships between temperament and eating behaviors in children [15,17]. Based on this knowledge, our study aimed to investigate the effect of lockdown measures on children’s eating behavior and if the changes were influenced by children’s nutritional status.

## 2. Materials and Methods

### 2.1. The Cohort

Children attending the Pediatric Clinic of the University of Campania “Luigi Vanvitelli” for nutritional or auxological counselling between January 2020 and February 2020 were enrolled. Parents were asked to answer a questionnaire about their children’s eating behavior at enrollment. The questionnaire was administered with the intention of assessing the behavioral phenotype that could affect weight-loss treatment compliance. As the COVID-19 lockdown did not allow us to complete the study as had been planned, we performed a second evaluation during lockdown containment to the verify potential effect of the lockdown on eating behavior phenotype in children. The ethics committee approved the study, and a signed informed consent was collected from each participant. Children affected by secondary obesity, food allergies, eating disorders, and not providing consent were excluded. Patients underwent clinical examination. Children were weighted undressed by a balance beam scale and height was measured by a Harpenden stadiometer. Z-scores for BMI were calculated by using the lambda–mu–sigma method according to reference values [18,19]. A BMI above the 85th and the 95th percentile for age and sex according to the WHO growth chart was used to define overweight and obesity [13], respectively.

Moreover, parents were asked to answer the Children’s Eating Behavior Questionnaire (CEBQ) at enrollment and at follow-up during lockdown containment (November 2020) through a phone call.

### 2.2. The Children’s Eating Behavior Questionnaire—CEBQ

The CEBQ is a brief, widely used, reliable, and validated self-report questionnaire which is easy to administer and interpret and addressed to parents. The questionnaire is designed to measure eating habits in children and for assessing early precursors of obesity or eating disorders. It has been shown to have good internal consistency, adequate test–retest reliability, and construct validity [12,13]. It consists of thirty-five items that evaluate eating habits on eight sub-scales: Food Responsiveness (4 items, FR), Enjoyment of Food (4 items, EF), Emotional Over-Eating (4 items, EOE), Desire to Drink (3 items, DD), Satiety Responsiveness (5 items, SR), Slowness in Eating (4 items, SE), Emotional Under-Eating (4 items, EUE), and Food Fussiness (7 items, FF). Caregivers rated the frequency of their child’s behaviors and experiences on a *five-point Liker scale*: 1 = never, 2 = rarely, 3 = sometimes, 4 = often, and 5 = always. The sum (or mean) for each sub-scale was calculated. The higher the score of each sub-scale, the more incidental the value of the eating behaviour was.

In addition to the eating style, the CEBQ incorporates sub-scales referring to appetite characteristics with potential links to obesity [20]. Previous studies have associated increased adiposity with scores for lower Satiety Responsiveness, Slowness in Eating, and increased Enjoyment of Food. These results support the idea that abnormal appetite traits are associated with weight variation in individuals [20].

### 2.3. Statistical Analysis 

Continuous variables were tested for normal distribution using the Kolmogorov and Smirnov test. Differences in individual scores between baseline and follow-up were investigated using the Wilcoxon paired-sample test as the continuous variables did not reach normal distribution because of the small sample size. Children were grouped according to BMI percentile in normal weight and overweight/obese group. Delta changes in CEBQ subscales were calculated and differences between the two groups were tested using the Mann–Whitney U test for independent samples. Data are expressed as median and interquartile range (IQR) or as frequency for categorical variables. The analyses were performed using SAS^®^ on Demand for Academics (SAS Institute Inc., Cary, NC, USA) software. *p* values < 0.05 were considered statistically significant.

### 2.4. Post-Hoc Power Calculation

This study derives from a post-hoc analysis of the data collected for nutritional and auxological evaluation in clinical practice. Therefore, the sample size was not established on the basis of the aims of the present study, and we performed a post-hoc power calculation. Based on 69 parents–children pairs that performed the two evaluation times, the power was >90% and the type I error was 0.05 with a calculated effect size of 0.3. The post-hoc power analysis was performed with G*Power version 3.1.9.6 (Heinrich-Heine-Universität Düsseldorf, Germany) [21].

## 3. Results

The final study sample includes a total of 69 children of both sexes (39 females and 30 males), aged 8.1 ± 4.1 years (Table 1).

At follow-up, we observed a significant increase in FR and EOE total scores (*p* = 0.009 and *p* = 0.001, respectively). Conversely, SR and DD subscales showed significantly lower levels during lockdown compared to baseline (*p* = 0.0001 and *p* = 0.04, respectively). No significant differences were found for FF, SE, and EF, whereas a trend toward a significant increase in EUE during lockdown was observed (*p* = 0.06) (Table 2).

Moreover, we investigated whether eating behavior changes differed between normal weight and overweight/obese groups. Delta changes in FR total scores were significantly higher in normal weight groups compared to overweight/obese groups (*p* = 0.02, Figure 1). No differences were found in the other CEBQ subscales.

## 4. Discussion

The COVID-19 pandemic containment measures have been responsible for lifestyle changes causing or worsening weight gain [22]. The lockdown also represented a stressful event for children, with a potential impact on their behavioral attitude and a drastic change in everyday life.

In our study, we sought to evaluate the changes in eating behavior during lockdown period in children. We observed significant changes in DD, EOE, FR, and SR subscales. DD, EOE, and FR subscales reflect the category of “interest in food”, whereas SR subscale belongs to the group of “no interest in food”. Previous studies reported that children eating behavior assessed by the CEBQ correlates with nutritional status, with BMI increasing with the increase of “interest in food” subscales scores [15]. In particular, DD reflects the desire to drink and has been associated with consumption of sugary drinks; therefore, it positively correlates with BMI [23]. In our study, we observed lower DD scores during lockdown. This finding is in contrast with the worsening of the other subscales; it might be hypothesized that “desire to drink” has been understood as “thirsty for water”. Children in the pandemic are likely to have been less thirsty when they stopped doing their usual activities.

FR reflects a greater susceptibility to external food stimuli, suggesting that home environment, aggravated by a forced closure and without social relations may have contributed to increasing the responsiveness to food. Studies using the CEBQ in comparison to BMI usually show that EOE is positively associated with BMI [24]. This subscale is characterized by an overeating in response to negative emotions. The positive correlation with an emotionally impactful period, such as the lockdown period, is easily understandable. Very often, the relationship that children establish with food will be carried on even in adulthood, by resorting to the emotional eating of “comfort food” in reaction to difficulties and negative emotions. Conversely, SR inversely correlates with BMI, with children with excess weight scoring lower on this sub-scale than normal weight peers [25]. A decreased response to satiety makes children less able to regulate food intake, and therefore contributes to excess weight gain. The important variation observed in our studies occurred as a result a greater influence, in the period observed, of the erroneous parenting practices towards these stimuli.

The COVID-19 pandemic had a massive impact on human health, causing changes in eating behavior [26]. Many studies show an increase in the consumption of sugary drinks, fatty foods, and a reduction in physical activity during the pandemic [5,6,9,27].

Our results agree with a recent French study that showed that increased boredom in children was strongly related to increased food responsiveness, increased emotional overeating, and increased snack frequency, even in young children [28]. Moreover, it should be considered that the COVID-19 pandemic has increased the exposure of children to family environment with major effect of parents feeding practices to children behavior. In fact, several sources of evidence report that parental feeding practices might favorite emotional eating in children, thus heightening the risk for overweight. Maternal use of food as reward, talking about food outside of mealtime, and parental-restrictive feedings are associated with increased weight gain during childhood [29,30,31]. The small sample size of our cohort limits the possibility to extend these findings to the general population. In addition, it should be acknowledged that our population derives from only one pediatric center and that the two questionnaires were submitted with different settings (face-to-face visit and phone call). However, the available literature on eating behavior changes during the COVID-19 pandemic in children is poor.

## 5. Conclusions

Our findings confirm that the environment in which we live and the influences to which we are subjected contribute to characterize the phenotype of eating behavior. In this context, lockdown containment has negatively impacted children’s eating behavior because of the forced restriction at home and social limitations. In particular, emotional eating, interest in food, and satiety regulation have shown significant changes in an obesogenic manner, affecting normal weight children more severely children compared to children with overweight and obesity. Over time, this phenomenon might lead to an increased prevalence of obesity, with a potential increase of new cases of overweight children and a worsening of the pre-existing ones. Therefore, prevention strategies for weight gain should be enhanced in order to counteract the negative effect of the lockdown on children’s eating behavior. However, these preliminary results should be more deeply investigated in larger cohorts. Additionally, longitudinal study might confirm this data with particular attention to obesity prevalence changes over time.

## Figures and Tables

**Figure 1 children-09-01078-f001:**
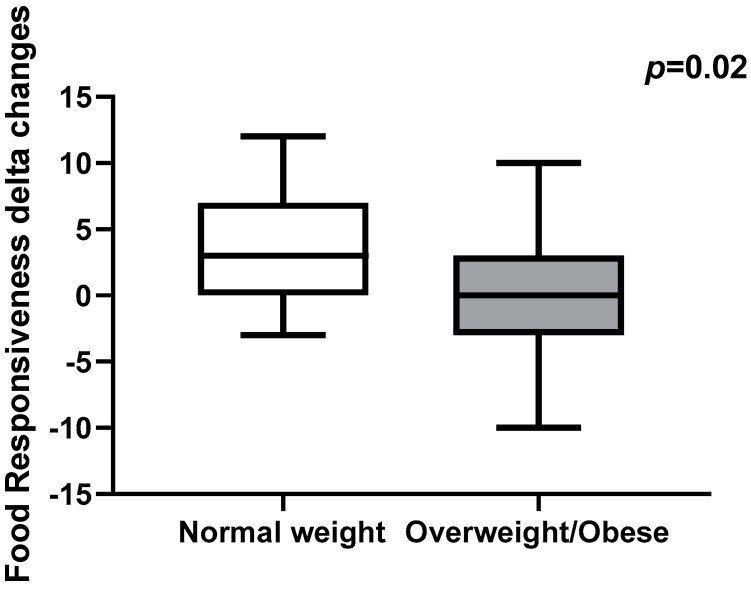
Delta changes in Food Responsiveness total scores in normal weight children and children with overweight and obesity. Data are expressed as median and interquartile range.

**Table 1 children-09-01078-t001:** Anthropometric and CEBQ scores of the study population.

Parameter	
Sex (M, %)	43.4
Age	7.1 (5.0–10.6)
BMI	20.5 (16.7–25.8)
BMI z-score	1.5 (0.7–3.0)
FR	11 (8–17)
EOE	8 (6–11)
EF	16 (13–18)
DD	7 (6–9)
SR	15 (12–17)
SE	10 (7–14)
EUE	10 (7–11)
FF	18 (13–23)

Data are expressed as median (interquartile range) or frequencies. Legend: BMI: body mass index; FR: food responsiveness; EOE: emotional over-eating; EF: enjoyment of food; DD: desire to drink; SR: satiety responsiveness; SE: slowness in eating; EUE: emotional under-eating; FF: food fussiness.

**Table 2 children-09-01078-t002:** Differences in CEBQ subscale before and during lockdown.

CEBQ Subscale	before Lockdown	during Lockdown	*p*
FR	11 (8–17)	13 (10–18)	**0.009**
EOE	8 (6–11)	10 (7–13)	**0.001**
EF	16 (13–18)	15 (12–18)	0.43
DD	7 (6–9)	6 (4–9)	**0.04**
SR	15 (12–17)	15 (12–17)	**0.0001**
SE	10 (7–14)	10 (8–12)	0.45
EUE	10 (7–11)	10 (8–13)	0.06
FF	18 (13–23)	17 (15–18)	0.17

Data are expressed as median (interquartile range). Statistically significant *p*-values are in bold. Legend: FR: food responsiveness; EOE: emotional over-eating; EF: enjoyment of food; DD: desire to drink; SR: satiety responsiveness; SE: slowness in eating; EUE: emotional under-eating; FF: food fussiness.

## Data Availability

Data are available upon motivated request to corresponding author.

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
