# Peer review of "Effect of COVID-19 Lockdown on Children’s Eating Behaviours: A Longitudinal Study"

_children, 2022, doi:10.3390/children9071078_

Round 1

Reviewer 1 Report

Dear Authors,

Thank you very much to Editor for inviting me to review your publication. Congratulations on your research on the field of children's eating behaviors. Improper eating habits can result in serious health consequences. This problem is particularly relevant in children.

The paper submitted for review, moreover, addresses the important issue of the impact of the lock down caused by the Covid-19 pandemic on changing children's eating behavior.

Below are my suggestions / comments:

The validated CEBQ scale used in the study is a valid tool for studying such behavior. However, a small sample (69 children) was used for the study. So it is difficult to talk about generalizing the results of the study.

Basic non-parametric statistical tests were used to measure differences (presumably because the normality of the variables' distributions was not confirmed).

The conclusion presented is very general and should be expanded to include results from the survey.

Author Response

We thank the Editor and reviewers for taking time in revise our manuscript and giving suggestions to improve it. We have addressed all the comments raised and changed the text according to them.

Reviewer’s 1 comments

  1. The validated CEBQ scale used in the study is a valid tool for studying such behavior. However, a small sample (69 children) was used for the study. So it is difficult to talk about generalizing the results of the study.

Answer: thank you for your suggestion, we have added it as study limitation, please see page 6 lines 209-218.

  1. Basic non-parametric statistical tests were used to measure differences (presumably because the normality of the variables' distributions was not confirmed).

Answer: thank you for your comment. We have specified the reason for this statistical text in methods section. Please see page 3 lines 114-115.

  1. The conclusion presented is very general and should be expanded to include results from the survey.

Answer: we agree with the reviewer and have changed the conclusions according to suggestions. Please, see page 6 lines 209-218.

Reviewer 2 Report

Dear Authors,

I consider your article "Effect of Covid-19 lockdown on children's eating behaviours: a longitudinal study" important to strengthen and expand the knowledge about the effects of lockdown measures taken worldwide to prevent the spread of the COVID-19 pandemic. The research instrument used to collect data, the CEBQ questionnaire is a standardized questionnaire, so I appreciate its usage because it offers to other researchers the opportunity to compare and analyse data.

Even so, in order to be published in this journal, I consider there are needed few improvements:

-       Introduction: could be enhanced, by adding also details regarding the importance of the aspects measured by the CEBQ questionnaire and also, example of the results from other researchers using CEBQ questionnaire

-       Research questions and methodology: there is mentioned only the general aim of this study, to see if there are any differences induced by the pandemic restrictions, but no other assumptions and hypotheses were made. Regarding the sample used for this research, it is an advantage the fact that the questionnaire could be applied before and during pandemic to the same sample, but the number of responses and especially the fact they are only clients of the Pediatric Clinic of University of Campania “Luigi Vanvitelli” is a strong limit of this research. In order to overcame it, in the conclusions part it should be at least mentioned this limitation, and also may be it was useful if the specificity of this sample was better explained (as it seems, they are individuals already interested in the “healthy eating subject”, or already having some affections or disease related to healthy eating habits).  

There are also other issues regarding the questionnaire and data collection methods, to be at least discussed, for example the errors that might be induced by the way the answers were collected F2F before pandemic, and by telephone after that.

-       Conclusions and discussions of the study should also be completed with some remarks regarding the limitations of your study and also, how can be further used your results by other researchers. Also, extended correlations could be made based on the scientific literature and your results, considering the characteristics of the families and their eating habits (because children eating behaviours are the result of the family habits).

Also, there are few mistakes that should be corrected (to make more clear the phrase from rows 68-72) and also the biography list (the way is numbered).

Generally speaking, after these corrections and additions, the article could be reviewed again and published, because, as I mentioned at the beginning, it is important to deepen the negative effects emerging during and after pandemic.

Author Response

We thank the Editor and reviewers for taking time in revise our manuscript and giving suggestions to improve it. We have addressed all the comments raised and changed the text according to them.

Reviewer’s 2 comments

  1. Introduction: could be enhanced, by adding also details regarding the importance of the aspects measured by the CEBQ questionnaire and also, example of the results from other researchers using CEBQ questionnaire

Answer: thank for your suggestion, we have included a brief description of CEBQ and related literature in introduction, please see page 2 line 58-67.

  1. Research questions and methodology: there is mentioned only the general aim of this study, to see if there are any differences induced by the pandemic restrictions, but no other assumptions and hypotheses were made. Regarding the sample used for this research, it is an advantage the fact that the questionnaire could be applied before and during pandemic to the same sample, but the number of responses and especially the fact they are only clients of the Pediatric Clinic of University of Campania “Luigi Vanvitelli” is a strong limit of this research. In order to overcame it, in the conclusions part it should be at least mentioned this limitation, and also may be it was useful if the specificity of this sample was better explained (as it seems, they are individuals already interested in the “healthy eating subject”, or already having some affections or disease related to healthy eating habits). There are also other issues regarding the questionnaire and data collection methods, to be at least discussed, for example the errors that might be induced by the way the answers were collected F2F before pandemic, and by telephone after that.

Answer: we thank the reviewer for raising these issues, we have added these suggestions as study limitations in conclusion section. Please see page 6 lines 209-218.

  1. Conclusions and discussions of the study should also be completed with some remarks regarding the limitations of your study and also, how can be further used your results by other researchers. Also, extended correlations could be made based on the scientific literature and your results, considering the characteristics of the families and their eating habits (because children eating behaviours are the result of the family habits).

Answer: we changed the conclusion according to suggestions and added a comment on family habits and children eating behaviors in discussion. Please, see page 6, lines 197-203, and page 6 lines 209-218.

  1. Also, there are few mistakes that should be corrected (to make more clear the phrase from rows 68-72) and also the biography list (the way is numbered).

Answer: thank you for the comment. We have rephrased this period; please, see page 2 lines 77-80. Moreover, we have modified bibliography.